

# Improvement of contact lens-associated dry eye disease with the use of hydrogen peroxide

Susana Castro[1], Laura Garcia-Aguilar[1], Eduardo Garcia-Brion[1], Sofia Pérez-García[1], Consuelo Rosique[1], Carmelo Baños[1,2] and Irene Sanchez[3,4]

[1] inGO Research Group, General Optica, Barcelona, Cataluña, Spain
[2] Departamento de Optometría y Visión, Universidad Complutense de Madrid, Madrid, Spain
[3] Departamento de Física Teórica Atómica y Óptica, Universidad de Valladolid, Valladolid, Castilla y Leon, Spain
[4] Optometry Research Group, IOBA Eye Institute, Valladolid, Castilla y Leon, Spain

## ABSTRACT

**Background**. The dropout rate of contact lens users has not decreased significantly over the years. Despite continuous improvements in contact lens (CL) designs, materials and surface treatments, the number of CL users who drop out remains similar to the number of new CL users. The aim of this study is to analyse the improvement in contact lens-associated dry eye disease (CLADE), quantified with the OSDI questionnaire when changing maintenance system solution from multipurpose solution to hydrogen peroxide.

**Methods**. This study included contact lens users for over a year as the multipurpose solution for the maintenance system, suffering from CLADE, and those who scored over 13 in the ocular surface disease index questionnaire, and did not manifest any clinical signs over 3 in the EFRON scale. The non-parametric data distribution was verified with the Kolmogorov-Smirnov test and Wilcoxon signed-rank test, which was used to compare the visual acuity (VA), OSDI score and bulbar redness (EFRON scale) of the follow-up visit against the baseline value.

**Results**. Thirty-eight patients were included. Analysing the clinical parameters between the initial and final visit after one month of hydrogen peroxide use, a statistically significant improvement was found in the VA, bulbar hyperemia, OSDI scale and their subscales of the total sample ($P < 0.04$).

**Conclusion**. This study is intended as a first step towards a standardised protocol of actions to improve CLADE in an attempt to reduce contact lens dropout using OSDI as a tool for detection.

Corresponding author
Irene Sanchez,
isanchezp@ioba.med.uva.es

## INTRODUCTION

The dropout rate of contact lens (CL) users has not decreased significantly over the years (*Marjorie, Mohinder & Marianne, 2014*). Despite continuous improvements in CL designs, materials and surface treatments, the number of CL users who drop out remains similar to the number of new CL users (*Pucker & Tichenor, 2020*). Analysing dropout

rates, 26% of new users drop out within the first year and nearly 50% in the first two months, ruling out problems of CL rejection due to overuse (*Sulley, Young & Hunt, 2017*). The causes of discontinued use are diverse, with the most common being contact lens discomfort (CLD) (24%–49%), which includes symptoms of burning or stinging. This is followed by dryness (9%–20%), blurred vision and redness (5%–11%). Other causes include the need to change the CL, discomfort with cleaning, discontinued use of professional advice, difficulty in handling, pregnancy, which together account for 49% of discontinued use (*Sulley, Young & Hunt, 2017*; *Pritchard, Fonn & Brazeau, 1999*; *Young et al., 2002*). CLD is defined by The Tear Film and Ocular Surface Society (TFOS) as a reduced compatibility between contact lenses and the ocular environment as a consequence of the splitting of the tear film by the contact lens (*Kojima, 2018*).

There are several possible causes of CLD that are being investigated as the origin of the problem although symptoms are known to worsen with the use of visual display terminal (*Sulley, Young & Hunt, 2017*; *Young, 2004*; *Alamri et al., 2022*). The use of CL may reduce the stability of the tear film and consequently increase evaporation, leading to contact lens-associated dry eye disease (CLADE) with symptoms as burning, redness, stinging or dryness especially by end-of-day pain wearing CL. In addition, CLD may result from the accumulation of deposits such as proteins/lipids on the CL surface affecting visual quality and poorer wetting of the CL leading to dehydration of the CL and lid wiper epitheliopathy (LWE) produced by the friction of the eyelid over the ocular surface when the tear film thins (*Kojima, 2018*; *Jadi et al., 2012*; *Garcia-Queiruga et al., 2024*). Moreover, silicone hydrogel material is known to have a tendency of adhering to protein accumulations/deposits on its surface, which is associated to problems of discomfort by activating inflammatory processes (*Young, 2004*). Corneal hypoxia, toxicity or mechanical damage, produces inflammatory mediators and cytokines in the corneal and conjunctival epithelium that induce hyperemia and neovascularization in CL users (*Alamri et al., 2022*; *Jadi et al., 2012*; *Garcia-Queiruga et al., 2024*).

The detection of this symptomology in early stages is key to avoid dropouts (*Jones et al., 2023*). Anamnesis is compulsory for detecting CLADE symptoms, including routine, activities and environmental conditions, although some users may perceive these symptoms as habitual in CL use. Standardised questionnaires should be helpful to quantify these symptoms in order to standardise actions that could be proposed (*Craig et al., 2017*). The ocular surface disease index (OSDI) questionnaire is validated as a useful tool to classify the degree of dry eye if it is combined with another clinical test. Moreover, it could be a tool for early detection and monitoring of CLADE and assess the outcomes of the actions to improve or resolve CLD and therefore the dropout rate of CL use (*Pastor-Zaplana et al., 2022*).

Proper CL cleaning is essential to prevent and remove lipid and protein deposits, ensure suitable wetting of the CL, and therefore the stability of the tear film and minimise possible future complications (*Muntz et al., 2015*). This is reported by TFOS, which in its 2024 report (*Jones et al., 2023*; *Garcia-Queiruga et al., 2024*), noted the relationship between the correct choice of a CL cleaning system and the comfort for users. It gives special relevance to surfactants as they can emulsify the lipid layer and destabilise the tear film. The TFOS

report also states that the use of surfactants is a key factor in the choice of a CL cleaning system (*Craig et al., 2013*).

Older studies showed hydrogen peroxide is a cleaning agent that has demonstrated several advantages over multipurpose CL care solutions (*Jones et al., 2023*). It is preservative-free, thereby avoiding hypersensitivity reactions that may cause discomfort for some patients (*Nichols et al., 2019*). It is more efficient at cleaning proteins and lipids deposited on the CL surface and its penetration of microbial biofilms, making it a promising option for those patients who are prone to the formation of such deposits, however, the use of peroxide is currently very limited (*Lievens et al., 2016*). For these reasons, the aim of this study is to analyse the improvement in CLADE, quantified with the OSDI questionnaire when changing the maintenance system solution from multipurpose solution to hydrogen peroxide.

## MATERIALS & METHODS

### Study design

A multicentric longitudinal, unmasked, prospective study has been carried out on patients attended at General Optica centres in the region of Castilla and Leon Community (Spain). Nine centres participated in this study. The professionals responsible for data collection were nine optometry graduates from these centers who follow the same examination protocols and routine practices enforced by clinical management since 2018. They have received specific training for data collection and result evaluation, provided by the principal investigator.

Forty patients (11 males and 29 females) were included but two female patients dropped out of the study because they did not show up for the final visit, 11 men and 27 women in the end. All patients use CLs for over a year, as the multipurpose solution for the maintenance system. Of which 26.32% used Biofinity (CooperVision, Pleasanton, CA, USA), 55.26% used Air Optix Aqua (Alcon Health Care, Fort Worth, TX, USA) and the remaining 18.42% used other monthly CL brands and the remaining 18.42% used other monthly CL brands, including Xtensa, Acuvue Oasys, Biomedics 55 Evolution, Saphir Rx Monthly, Purevision 2 HD, Acuvue Vita, and Gentle 59.

These patients suffering from CLADE (*Alamri et al., 2022*) defines as feeling symptoms such as dryness, sensing a foreign body, eye strain, and blurred vision that worsen throughout the day, even pain at the end-of-day wearing CL's, and those who scored over 13 in the OSDI questionnaire, and did not manifest any clinical sign over 3 in all signs included in the EFRON scale. All patients kept the same CL and refractive error prescription in both visits with the maintenance system being the only change allowed. All those patients who showed systemic or eye pathology, ocular surgery, vision impairment, pregnant or breastfeeding mothers or environmental allergies were excluded (*Efron, 1998*).

A minimum sample size of 32 subjects was determined to be necessary to detect a minimum difference of 6 points measured with OSDI score in different visits with an alpha risk of 0.05 and a beta risk of 0.1, assuming a standard deviation of 10 points in OSDI score. We finally included 38 volunteers to guarantee an adequate sample size for

statistical analysis even if 20 per cent of the subjects dropped out of the study. Moreover, we performed different grouping within the total sample because assuming an alpha risk of 0.10 and a beta risk of 0.2, a sample size of 19 patients will be enough. In this way, data were compared by gender, age, years of use, hours of use per day, time since CLADE was perceived and type of contact lens.

The study was approved by the Human Science Ethics Committee (PI 21-2421) of Valladolid East-Area (Hospital Clinic, Castilla y Leon Public Health System-SACYL). During the initial visit, all participants received all the information related to the study and an informed consent was obtained prior to any clinical procedure. All patients were treated in compliance with the Declaration of Helsinki and the European Normative for data protection (2016/679 Regulation of the European Parliament and the Council of 27th of April 2016, General Data Protection Regulation).

## Procedure

A complete eye-exam was performed to ensure patients check the inclusion criteria. The eye exam included measurements of visual acuity (VA) with contact lenses, subjective refraction if it was necessary, a detailed anterior segment evaluation using biomicroscopy, graded according to the EFRON scale, and completion of the OSDI questionnaire. Participants were recruited from among those attending their routine contact lens check-ups at General Optica centers. Individuals who reported the previously described symptoms during these visits, and who met the inclusion criteria and none of the exclusion criteria, were invited to participate in the study. Clinical data of age, gender, visual acuity (VA) and refractive error, years using CL, hours per day, days per week and slit lamp bulbar hyperemia (EFRON scale). OSDI questionnaire were collected in the initial visit and follow-up visit after a month of using hydrogen peroxide solution (Disco, Disop, Spain). Both eye-exams were compared to analyse these variables in order to find an improvement with the use of hydrogen peroxide.

The OSDI questionnaire has twelve questions grouped into three categories or subscales: ocular symptoms (O. symptoms), vision related function (VRF) and environmental triggers (Envir. triggers). It offers five possible answers (0: none of the time; 1: some of the time, 2: half of the time, 3: most of the time, 4: all of the time; and not available). A total of subscale score is obtained with the addition of all answers times 25 and divided by the number of questions answered. This questionnaire classifies the dryness level the patient suffers. The patients are considered symptomatic from 13 points or above. It differentiates from normal (0–12 points), mild (13–22 points) to moderate (23–32 points) and severe dry-eye disease (33–100 points). Moreover, the OSDI questionnaire demonstrated to have sensitivity and specificity in distinguishing between normal patients and patients with dry eye disease, in combination with other trials (*Schiffman et al., 2000*; *Nair et al., 2018*).

The initial visit was performed at the first consultation, regardless of which day of the CL use cycle they were on, the second approximately one month later, again without disturbing the subject's normal replacement cycle. The EFRON scale was used as reference by different optometrists in this multicentric study. It consists of a clinical sign scale from zero to four. A value of 3 or 4 would imply an eye with compatible signs of ocular pathology.

In this case, the patients would be excluded from the study and advised to take a temporal discontinuation of CL use and referred to the ophthalmologist (*Efron, 1998*; *Pult, Purslow & Murphy, 2011*).

## Data analysis

Statistical analysis was performed using the SPSS 23.0 (SPSS, Chicago, IL, USA) statistical package for Windows. The non-parametric data distribution was verified with the Kolmogorov–Smirnov test ($P < 0.05$ indicated that the data were non-parametrically distributed). The results are presented as the mean ± standard deviation (SD) and range (minimum-maximum). The Wilcoxon signed-rank test was used to compare the VA, OSDI and subscales score, bulbar redness (EFRON scale score) of the follow-up visit (using hydrogen peroxide for a month) against the initial visit value (with their habitual multipurpose solution). Moreover, Cronbach's alpha coefficient was used to evaluate the internal consistency of OSDI questionnaire and their sub-scales. An alpha coefficient of 0.80 or higher was considered as an acceptable threshold for reliability.

The variations in these clinical parameters were analysed depending on the gender, the age, the years of CL use, years of CLADE manifestation, hours the use per day and CL design (spheric, toric or multifocal). The homogeneity of these groups was analysed with the U of Mann–Whitney ($P < 0.05$ was considered significant) in order to find the reason for the different improvements in the variables studied. Besides, bulbar hyperemia was also analysed as a dichotomous variable using cross-tabs and Fisher's exact test.

## RESULTS

### Comparison between each created group

The data analysed took into account gender, age, hours of use per day, time since using CL, time since noticing CLADE and type of CL. Descriptive data for each group are shown in Table 1. The only statistically significant difference ($P = 0.01$) between males and females is bulbar hyperemia during the final visit ($0.82 \pm 0.40$ *versus* $0.30 \pm 0.46$ respectively).

It has been observed that among those who have been using CL over a 16-year period and those who have been using it under that period of time, there are statistically significant differences ($P < 0.04$) in age ($44.55 \pm 8.17$ years *versus* $32.30 \pm 10.56$ years), hours of use per day ($6.66 \pm 0.68$ h *versus* $5.75 \pm 1.29$ h) and final OSDI score ($5.72 \pm 4.04$ *versus* $9.50 \pm 8.12$).

Statistically significant differences ($P < 0.03$) between users of less than 8 h per day and users of more than 8 h per day are found in the initial OSDI score ($16.60 \pm 6.25$ *versus* $11.76 \pm 3.94$) and final OSDI environmental triggers subscale score ($26.00 \pm 19.48$ *versus* $12.15 \pm 14.81$).

Regarding lens geometry (spherical, toric and multifocal), statistically significant differences ($P \leq 0.01$) are found in age ($37.07 \pm 13.04$; $34.44 \pm 8.62$; $49.42 \pm 6.07$ years respectively) and initial VA of the left eye ($0.97 \pm 0.07$; $0.88 \pm 0.14$; $0.91 \pm 0.09$).

Analysing patients according to the length of time they had been experiencing discomfort, less than one year or more, statistically significant differences ($P = 0.02$) were observed in the final score of the OSDI VFR subscales ($12.27 \pm 10.66$ *versus* $27.22 \pm 23.07$).

Castro et al. (2024), *PeerJ*, DOI 10.7717/peerj.18482

**Table 1  Summary of the descriptive parameters of the total sample and by groups.** The mean ± standard deviation is presented together with the range (minimum–maximum).

| | Total n = 38 | Men n = 11 | Women n = 27 | p | >40 years n = 19 | <40 years n = 19 | P | Using CL >16 years n = 18 | Using CL <16 years n = 20 | p | CL use until 8hours/day n = 25 | CL use more than 8hours/day n = 13 | p | Spherical CL n = 13 | Toric CL n = 18 | Multifocal CL n = 7 | P | Discomfort <1 year n = 16 | Discomfort >1 year n = 22 | p |
|---|---|---|---|---|---|---|---|---|---|---|---|---|---|---|---|---|---|---|---|---|
| Age (years) | 38.1 ± 11.2 (21 to 62) | 42.9 ± 13.6 (22 to 62) | 36.1 ± 9.7 (21 to 52) | 0.24 | 47.8 ± 5.9 (40 to 62) | 28.3 ± 4.6 (21 to 40) | **<0.01** | 44.5 ± 8.1 (28 to 62) | 32.3 ± 10.5 (21 to 62) | **<0.01** | 37.4 ± 9.3 (22 to 52) | 39.3 ± 14.5 (21 to 62) | 0.95 | 37.0 ± 13.0 (21 to 62) | 34.4 ± 8.6 (22 to 50) | 49.4 ± 6.0 (44 to 62) | **0.01** | 38.3 ± 11.9 (22 to 62) | 37.9 ± 11.0 (21 to 62) | 0.80 |
| Sph RE (Diopters) | −2.92 ± 3.00 (−10 to 5) | −2.57 ± 3.25 (−7 to 5) | −3.06 ± 2.94 (−10 to 3) | 0.90 | −2.78 ± 2.42 (−8 to 2) | −3.07 ± 3.54 (−10 to 5) | 0.73 | −3.42 ± 2.97 (−10 to 3) | −2.48 ± 3.02 (−7 to 5) | 0.38 | −3.10 ± 3.29 (−10 to 5) | −2.58 ± 2.41 (−8 to 2) | 0.43 | −3.27 ± 2.66 (−8 to 2) | −2.93 ± 3.61 (−10 to 5) | −2.25 ± 1.90 (−5 to 1) | 0.37 | −2.06 ± 2.56 (−7 to 3) | −3.55 ± 3.19 (−10 to5) | 0.06 |
| Sph LE (Diopters) | 3.05 ± 2.60 (−8 to 5) | −2.75 ± 2.96 (−7 to 5) | −3.18 ± 2.48 (−8 to 2) | 0.97 | −3.28 ± 1.80 (−7 to 1) | −2.83 ± 3.23 (−8 to 5) | 0.82 | −3.54 ± 2.15 (−8 to 2) | −2.61 ± 2.92 (−7 to 5) | 0.36 | −3.02 ± 2.99 (−8 to 5) | −3.12 ± 1.69 (−6 to 1) | 0.95 | −3.69 ± 1.72 (−7 to −1) | −3.04 ± 3.28 (−8 to 5) | −1.89 ± 1.60 (−3 to 1) | 0.44 | −2.53 ± 2.59 (−7 to2) | −3.43 ± 2.59 (−8 to5) | 0.20 |
| Hours/day | 8.34 ± 2.76 (2 to 16) | 9.18 ± 3.45 (2 to 14) | 8.00 ± 2.41 (2 to 16) | 0.25 | 8.94 ± 3.06 (2 to 16) | 7.73 ± 2.35 (2 to 13) | 0.27 | 8.33 ± 3.21 (2 to 16) | 8.35 ± 2.36 (4 to 14) | 0.97 | 7.00 ± 1.82 (2 to 8) | 10.92 ± 2.43 (9 to 16) | **<0.01** | 9.61 ± 2.56 (7 to 16) | 7.33 ± 2.72 (2 to 13) | 8.57 ± 2.50 (6 to 14) | 0.11 | 8.18 ± 2.81 (2 to14) | 8.45 ± 2.78 (2 to 16) | 0.57 |
| day/week | 6.18 ± 1.13 (3 to 7) | 6.45 ± 0.93 (5 to 7) | 6.07 ± 1.20 (3 to 7) | 0.37 | 6.57 ± 0.69 (5 to 7) | 5.78 ± 1.35 (3 to 7) | 0.10 | 6.66 ± 0.68 (5 to 7) | 5.75 ± 1.29 (3 to 7) | **0.03** | 6.12 ± 1.20 (3 to 7) | 6.30 ± 1.03 (4 to 7) | 0.74 | 6.30 ± 0.94 (5 to 7) | 5.88 ± 1.36 (3 to 7) | 6.71 ± 0.48 (6 to 7) | 0.29 | 6.12 ± 1.02 (4 to 7) | 6.22 ± 1.23 (3 to 7) | 0.47 |

**Notes.**

Sph, sphere; RE, right eye; LE, left eye; CL, Contact Lens.

Age category (18 to 62 years); range of years using CL (1 to 23 years); range of hours of CL wear per day (2 to 16 h per day); years of discomfort with contact lenses (1 to 5 years); range of days of CL wear per week (3 to 7 days per week). Wilcoxon rank test. Statistically significant values (p-value) are marked in bold.

The reliability of OSDI questionnaire was 0.87 (measured using the Cronbach's alpha) and their subscales were 0.79 for VRF, 0.76 for the O. symptoms and 0.65 for Envir. triggers.

**Intersession outcomes**

All parameters measurement showed an improvement statistically significant after the use of hydrogen peroxide (visual acuity, bulbar hyperemia and the OSDI questionnaire and its subscales, $p < 0.04$). Analysing the clinical parameters between the initial visit and final visit after one month of hydrogen peroxide use (Fig. 1), a statistically significant improvement was found in the VA of the total sample ($P < 0.04$), in the group of women ($P < 0.03$), in people who have been using CL for more than 16 years ($P = 0.02$), in people who use CL less than 8 h a day ($P < 0.04$) and in people who have felt discomfort for less than one year ($P = 0.03$). Also, in the VA of the left eye of toric CL users ($P = 0.04$).

Regarding bulbar hyperemia (EFRON scale), statistically significant improvement was found after hydrogen peroxide use for one month in all groups analysed except in men and in the multifocal CL group ($P > 0.16$) as shown in Fig. 2. However, when hyperemia is analysed as a dichotomous variable (considering the zero value of the EFRON scale as healthy and all others as impaired) with cross-tabs with Fisher's exact test the statistical significance disappears ($p = 0.08$) although the bulbar hyperemia disappeared in 16 patients.

The OSDI total score (Fig. 3) of all groups studied show a statistically significant improvement. In the OSDI subscales (Fig. 4), the VRF subscale and the environmental triggers subscale improved after one month of hydrogen peroxide use in all groups ($P < 0.04$). However, the ocular symptoms subscale did not improve significantly ($P > 0.21$) in the group of patients under 40 years of age, patients who have been using CL for less than 16 years and the multifocal CL users who have been experiencing discomfort for over a year.

## DISCUSSION

Despite improvements in CL materials and design in recent years, the number of dropouts among CL users has not decreased significantly (*Pucker & Tichenor, 2020*). For this reason, it is necessary to approach the problem from a different perspective. Several studies have focused on the importance of cleanliness in CL for reduce CLD in users with CLADE or similar symptoms (burning, stinging, dryness) which according to studies is one of the most reported causes (24%–49%) (*Sulley, Young & Hunt, 2017*; *Pritchard, Fonn & Brazeau, 1999*; *Young et al., 2002*; *Kojima, 2018*), without reaching standardised clinical protocols to solve the problem. Hydrogen peroxide is a well-studied alternative, with positive scientific evidence on the use of this solution and the improvement of comfort for CL users but without application in clinical practice (*Jones et al., 2023*; *Nichols et al., 2019*; *Lievens et al., 2016*; *Keir et al., 2010*; *Guillon et al., 2015*; *Moro et al., 2018*). This cleaning regime was more widely used in the 1980s but has since become an exceptional method, replaced by multipurpose solution. However, studies over the years have shown its advantages regardless of the evolution of contact lens materials. Moreover, it is again presented as a

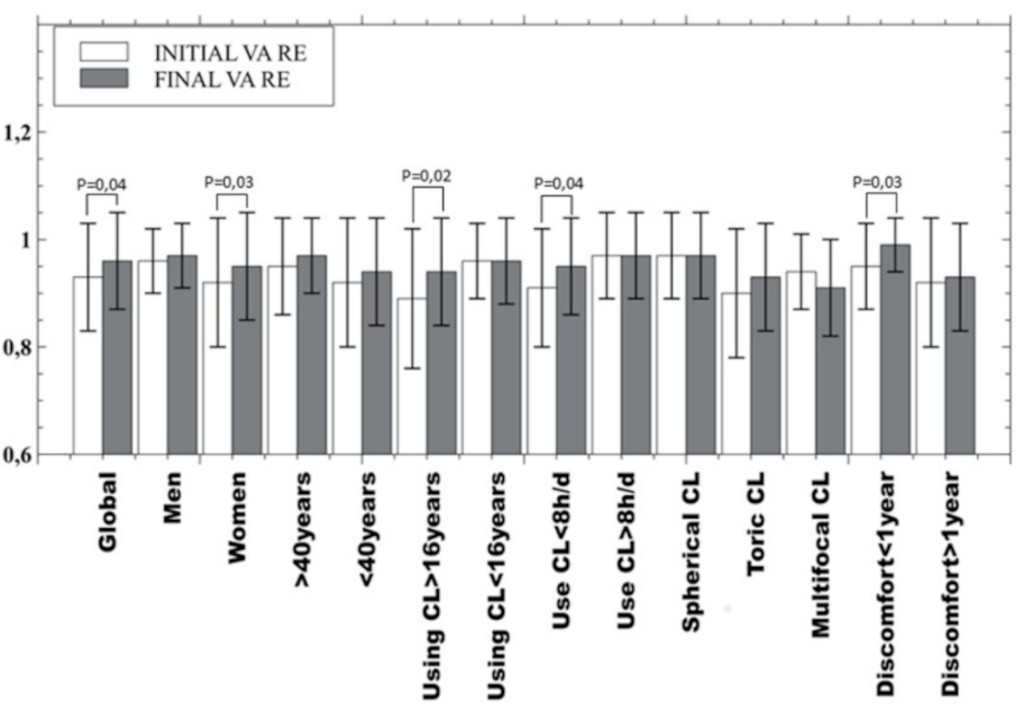

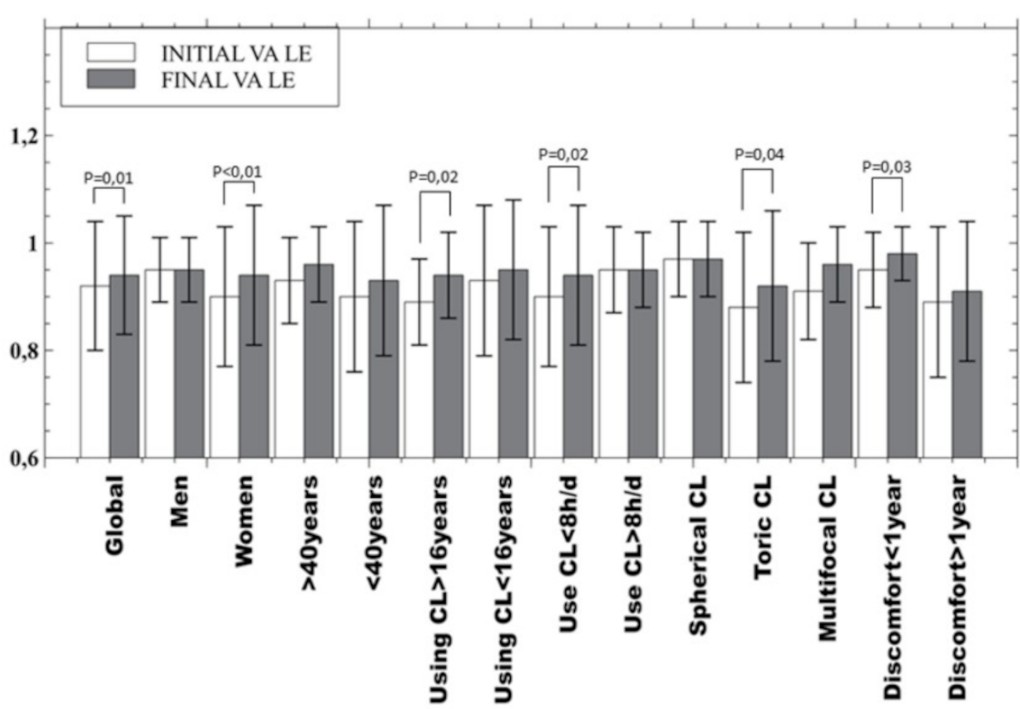

**Figure 1** **Mean and standard deviation of visual acuity values, overall result and in the subgroups analysed.** Top: right eye and bottom: left eye. Only *p*-values below statistical significance are included. CL, contact lens; h, hours; d, day; VA, Visual acuity; RE, right eye; LE, left eye.

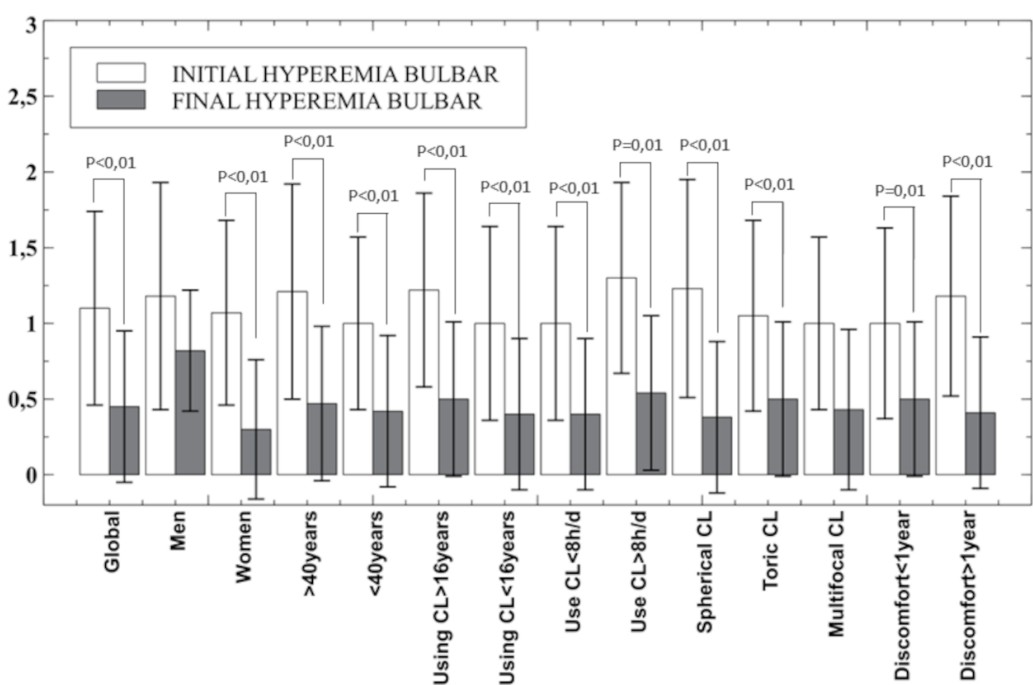

**Figure 2 Mean and standard deviation of bulbar hyperemia overall result and in the subgroups analysed.** Only *p*-values below statistical significance are included. CL, contact lens; h, hours; d, day.

high priority option because it has proven to be a disinfectant with higher efficacy than multipurpose solution, even on microbial biofilms (*Jones et al., 2023*; *Nichols et al., 2019*).

In this study, in order to test the effect of only changing the cleaning system (from multipurpose solution to hydrogen peroxide), subjects continued to wear their usual CL in the same way as before, *i.e.,* same CL, same refraction, same hours and same replacement. This aspect has been extensively studied; however, this study includes an easy clinical tool, the OSDI questionnaire, for patient classification (*Nichols et al., 2019*; *Lievens et al., 2016*; *Keir et al., 2010*; *Guillon et al., 2015*; *Tichenor et al., 2021*). The results obtained are congruent with other studies that were analysed in users of monthly contact lenses. The change from multipurpose solution to hydrogen peroxide (between 1 and 3 months) provides greater efficiency in removing deposits, obtaining 86% for mucus and 87% in lipids; implying a more wettable CL surface and therefore less desiccation of the LC expressed by pre-lens tear breakage time through Tearscope, decreased (7%), as a corneal staining (including hydrogel silicone CL); improved palpebral tissue integrity in the 50% of patients, thus less likelihood of the patient manifesting CLADE (*Pritchard, Fonn & Brazeau, 1999*; *Nichols et al., 2019*; *Lievens et al., 2016*; *Guillon et al., 2015*).

This improvement can be related to the clinical parameters measured in this study, as less desiccation is compatible with the improvement found in the OSDI score results by decreasing the mean score by almost half (14.94 to 7.71 points), a general decrease in bulbar hyperemia [except in the case of men ($P = 0.16$) and in multifocal CL ($P = 0.1$)] and a slight improvement in visual acuity is statistically significant ($P < 0.04$) as shown in

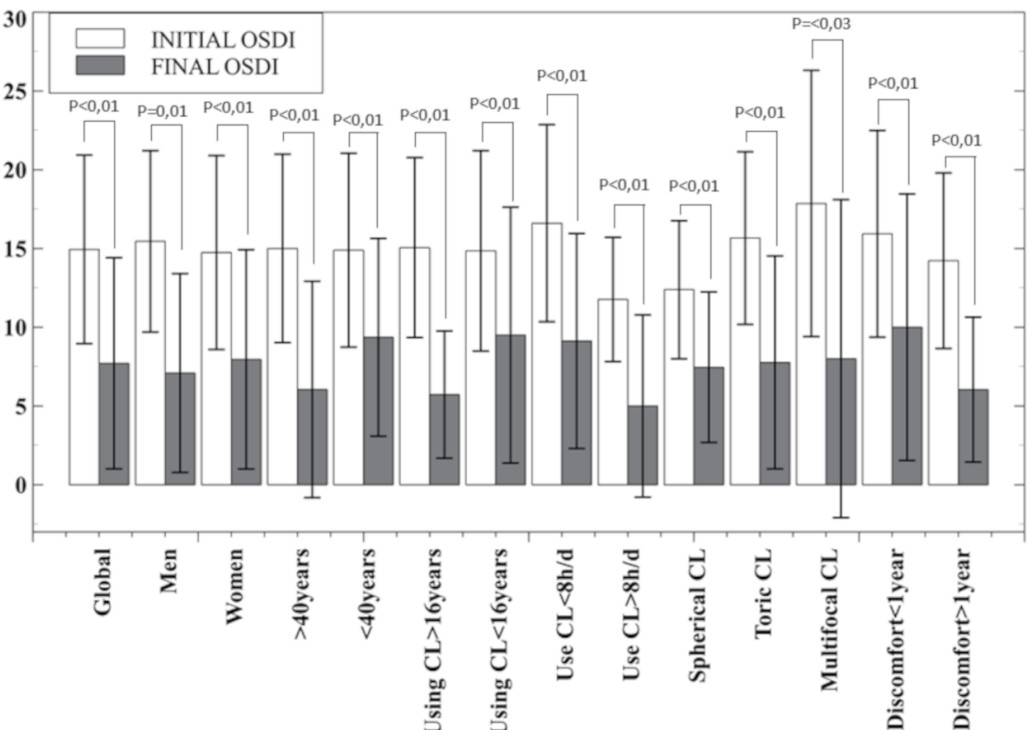

**Figure 3** **Mean and standard deviation of OSDI score overall result and in the subgroups analysed.** *P*-values below statistical significance are included. CL, contact lens; h, hours; d, day.

Fig. 1, in the global results and women, CL users for more than 16 years, CL users during less than 8 h per day and CL users with discomfort over a year. The OSDI questionnaire has proven to be a useful tool in the detection and monitoring of CLD or CLADE. For this reason, it would be reasonable to use the OSDI questionnaire in the fitting process of CL in order to avoid the dropout in the first months of use (*Sulley, Young & Hunt, 2017*), even in annual revisions.

Other questionnaires have been used for this purpose. The patients recruited by Kadence International (Boston, MA, USA) completed a questionnaire with a standard 6-point Likert scale of agreement or disagreement to assess changes in patients' symptoms and their intention to discontinue CL use after changing their habitual multipurpose solution to hydrogen peroxide. The follow-up survey was conducted six months later to determine that 93% of the patients continued with the hydrogen peroxide as a cleaning system and used the CLs at least once a week and the other 7% did not use CLs after 6 months of the initial visit (*Marjorie, Mohinder & Marianne, 2014*).

Not all studies have had positive results in the use of hydrogen peroxide. Keir et al. found no statistically significant differences comparing hydrogen peroxide (AO Sept Plus, Alcon EEUU) *versus* the multipurpose solution (OPTI-FREE; Alcon, Forth Worth, TX, USA) (*Keir et al., 2010*). In the same way, *Moro et al. (2018)* included the use of artificial drops together with hydrogen peroxide as a cleansing regime and found a statistically significant improvement *versus* multipurpose solution in conjunctival hyperemia, through

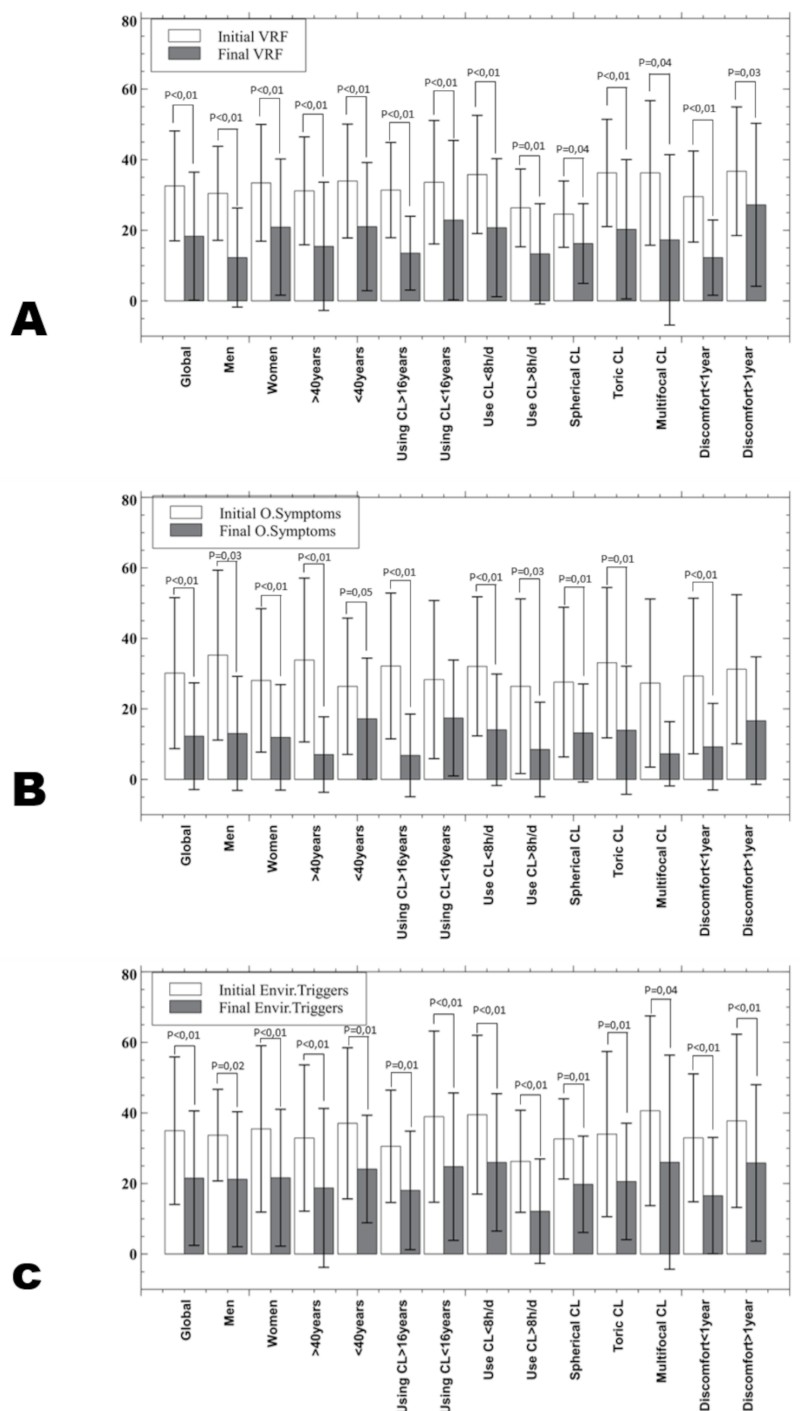

**Figure 4 Mean and standard deviation of the OSDI subscales in all the groups studied.** (A) vision related function (VRF); (B) Ocular symptoms (O. symptoms); (C) Environment triggers (Envir. triggers). Only *p*-values below statistical significance are included. CL, contact lens; h, hours; d, day.

a method similar to the EFFRON scale used in this study and tear stability measured with NIBUT and BUT. In this study, only men and multifocal CL users did not show statistically significant improvement in bulbar hyperemia ($P > 0.10$) as the ocular symptoms subscale of OSDI questionnaire in under 40 years old CL users, CL users for more than 16 years, multifocal CL users and CL users who feel discomfort for more than a year ago. It should be noted that not all clinical signs or symptoms are related to deposits or lack of wettability in the CL. Limbal hyperemia may indicate corneal hypoxia either by the mechanical effect of the lens and it has been previously shown that eyes wearing hydrogel silicone CL are less likely to show increased limbal redness, although it has not been found between lens material and dropout rate. Nevertheless, there seems to be a consensus that improving lens cleanliness improves wearer comfort (*Papas et al., 1997*).

Analysing the different groupings made in this study, some did not show statistically significant ($P > 0.06$) improvement in VA (men, CL users for more than 16 years, CL users for more than 8 h per day, multifocal and spherical CL users and CL users who feel discomfort for more than a year ago). Nevertheless, *Pritchard, Fonn & Brazeau (1999)* found in their study that some CL users (4%) drop out because of poor vision (caused by tear film components, such as lipids, proteins and mucins, accumulate on the CL surface), a parameter in which improvement has been found in few groups of this study (Fig. 1) (*Pritchard, Fonn & Brazeau, 1999*; *Willcox et al., 2021*; *Luensmann & Jones, 2008*).

These results suggest that some patients have CLD problems for which the hydrogen peroxide may not be the solution, although further studies are needed to corroborate this hypothesis. Besides, the rate of complications associated with CL use is not related with a cleaning regime according to a retrospective study comparing two cohorts of 3–5 year old CL users (one with hydrogen peroxide and one with multipurpose solution), although clinically it appears that the use of hydrogen peroxide is associated with a lower rate of giant papillary conjunctivitis and microbial keratitis, but the authors attribute this outcome to the heterogeneity of the samples compared (*Tichenor et al., 2021*).

However, there is no work that has proposed a protocol on how or when to use hydrogen peroxide to solve CLADE or CLD. This study shows that changing the cleaning regime from multipurpose solution to hydrogen peroxide may be the first option when starting to experience CLADE especially if the symptoms have been present for less than a year. Further studies are needed to determine different actions in those who have not improved, as is the case of men or multifocal CL. In addition, it is necessary to focus on complementary actions, such as the use of artificial drops or other recommendations in those patients whose rate of improvement on comfort was lower than other authors recommended (*Lievens et al., 2016*; *Guillon et al., 2018*).

## Study limitations

This study has some limitations such as the small sample size of some groups (men or multifocal CL users), which motivates the limitation of the significance of statistical analysis and the lack of a control group. Although these subgroups have been useful to compare if all patients have a proportional improvement and to propose which is the patient profile that would improve with this change of cleaning regime, which is an important

step in standardising the recommendation. The other limitation of this study is that the follow-up was only one month. It may be that some patients could continue to decrease due to inflammation issues, less palpebral papillae marking, or palpebral hyperemia as shown in other studies with a follow-up of 3 months, but in the first month, there is an improvement in all the parameters analysed (*Pastor-Zaplana et al., 2022*; *Tichenor et al., 2021*). Additionally, the study did not control for the day of the contact lens use cycle during the initial consultation. This may have influenced the results, as lenses examined toward the end of their use cycle could have accumulated more deposits and exhibited greater deterioration, introducing potential bias. Future studies should consider standardizing the day of the lens cycle for initial assessments to minimize this bias although the results have been very positive despite not taking this fact into account.

## CONCLUSIONS

In conclusion, this study is proposed as a first step towards a standardised protocol of actions to improve CLADE in an attempt to reduce CL dropout using the OSDI questionnaire as a tool for its detection and monitoring. Changing the cleaning regime from a multipurpose solution to hydrogen peroxide would be a simple and effective action for all those with an OSDI score superior to 13 points; especially in women, CL users for more than 16 years, CL users for less than 8 h a day and those who have reported CLADE for less than a year, as the improvement in these groups is statistically significant in all the parameters studied (VA, hyperemia, OSDI questionnaire and its subscales).

## ACKNOWLEDGEMENTS

The authors would like to thank General Optica centres staff of Burgos, Gamonal, Las Arenas, Las Palmas, León, Mantería, Palencia, Plaza España, Ponferrada, Pulido, Salamanca, Segovia, Tenerife, Valladolid and Zamora for data collection. Special thanks to the inGO research group (Abril-Frontela D., Aldeguer-Duran A, Alonso-Gil V, Álvarez-Alonso SM, Andrés-Álvarez V, Aramendia-Olaverri M, Ballesteros-Hernando A, Baranda-Pastor MA, Barriuso-Galván FJ, Benito-Fernández-Campo E, Burón-Reguera F, Cabrero-Antoranz A, Calvo-Ruso A, Carrera-García FJ, Castro-Medina MT, Contreras-Rios E, Coria-Cancelo G, Diez-Mocha H, Escudero-Aroca A, Fraile-García E, Gallego-Andrés O, Gallegos-Cocho I, García-Ablanedo MJ, García-García B, García-Juárez M, García-Mulas C, Heras-García S, José-Castro G, Lajarín-Navarro N, Ledesma-Gutiérrez L, León-Melendre C, López-González J, López-Infante JM, Lorenzo-García C, Marciel-Cimarra E, Martín-Arribas M, Matesanz-Rio F, Mediavilla-Mielgo C, Palacios-Val A, Pérez-García ML, Pescador-Martín T, Pinto A, Ribot-Jiménez I, Saiz-Montolio AM, Santamaría-López T, Vila-Alonso S, Villacampa-Murcia M).

### Funding

The authors received no funding for this work.

## Competing Interests

Susana Castro, Laura Garcia-Aguilar, Eduardo Garcia-Brion, Sofia Perez-Garcia, Consuelo Rosique and Carmelo Baños are employed by General Optica.

## Author Contributions

- Susana Castro performed the experiments, authored or reviewed drafts of the article, and approved the final draft.
- Laura Garcia-Aguilar performed the experiments, prepared figures and/or tables, authored or reviewed drafts of the article, and approved the final draft.
- Eduardo Garcia-Brion performed the experiments, authored or reviewed drafts of the article, and approved the final draft.
- Sofia Pérez-García performed the experiments, authored or reviewed drafts of the article, and approved the final draft.
- Consuelo Rosique performed the experiments, analyzed the data, authored or reviewed drafts of the article, and approved the final draft.
- Carmelo Baños conceived and designed the experiments, analyzed the data, prepared figures and/or tables, authored or reviewed drafts of the article, and approved the final draft.
- Irene Sanchez conceived and designed the experiments, analyzed the data, prepared figures and/or tables, authored or reviewed drafts of the article, and approved the final draft.

## Human Ethics

The following information was supplied relating to ethical approvals (*i.e.*, approving body and any reference numbers):

The study was approved by the Human Science Ethics Committee (PI 21-2421) of Valladolid East-Area (Hospital Clinic, Castilla y Leon Public Health System-SACYL).

## Ethics

The following information was supplied relating to ethical approvals (*i.e.*, approving body and any reference numbers):

Human Science Ethics Committee of Valladolid East-Area (Hospital Clinic, Castilla y Leon Public Health System-SACYL)

## Data Availability

The main patient variables analyzed in this study are available in the Supplemental Files.

## Supplemental Information

Supplemental information for this article can be found online at http://dx.doi.org/10.7717/peerj.18482#supplemental-information.

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
