# Peer review of "Improvement of contact lens-associated dry eye disease with the use of hydrogen peroxide"

_PeerJ, doi:10.7717/peerj.18482_

## Round 0.1 · original submission · Major Revisions

The manuscript has been reviewed by three experts in the field. Revisions are
necessary before the manuscript is suitable for publication

Reviewer 1 ·

Basic reporting

I consider that the article complies with what has been indicated.

Experimental design

From my perspective, it has been a useful topic for the contact lens clinic. However, some contributions and comments could be made to improve the article, especially in the materials and methods:

- In the methodology you have used the OSDI questionnaire in LC user subjects, why have you not also considered using the “Contact Lens Dry Eye Questionnaire CLDEQ-8” questionnaire?

- Line 123: “A complete eye-exam was performed to ensure patients check the inclusion criterio”. The authors should specify in more detail the complete eye-exam and the inclusion criteria, how the participants were recruited and selected, and whether the tests were performed with or without LC.


- Line 143: “The initial test was performed at the first consultation, regardless of which day of the CL use cycle they were on,…”. From my point of view, in this study of monthly contact lenses, it is very important to take into account the LC use cycl. The fact starting the first visit testing with the LC on day 29 of use, may be causing worse results due to deposit accumulation and deterioration. These results are different from the user who started the study on day 1 of the cycle. Failure to control for this in the different participants causes a bias and limitation in the study.

- How many professionals and centers participated in the study? Were possible differences between them taken into account? Particularly in the subjective analysis of the bulbar redness (EFRON scale).

- During the month of study, there were no complications with the use of hydrogen peroxide, nor loss of the sample?

- For the statistical analysis, which eye was chosen for the statistical analysis? Was it random?

- Line 172: “18.42% used other monthly CL brands”. They should indicate which other brands or materials.

- Table 1. Missing units of measurement (years, diopters...). What is the maximum and minimum value within the “Using CL > 16 years group” and “Using CL < 16 years group”? This should be described in the table.

- Line 233: “The change from multipurpose solution to hydrogen peroxide (between 1 and 3 months) provides greater efficiency in removing deposits”. Why have the deposits not been analyzed?

Validity of the findings

No comment

Reviewer 2 ·

Basic reporting

1. Line 30: Please use the acronyms properly. In methods section of the abstracts the authors should change “contact lens users” to “CL users”.
2. Line 31 – 32: Add OSDI after “in the ocular surface disease index (OSDI) questionnaire”
3. The authors should state which “clinical signs” were studied in the abstract. Did the authors only measured the bulbar redness as they show later?
4. The authors should clearly state in the methods section of the abstract that participants were examined in two sessions (baseline and follow-up). It is not clear.
5. Please be consistent with how you refer to OSDI (Line 34: OSDI score and Line 38: OSDI scale). Revise.
6. Line 38: “subscales” of what?
7. Please revise the line 49 to 54 because there are too many ideas and is difficult to understand. Perhaps if you separate it into two sentences it will be clearer.
8. I think the authors should introduce the concept of Contact Lens Discomfort (CLD), which always appears together with dry eye symptoms, but these symptoms disappear once the contact lenses are removed. Please elaborate on this and link it with the concept of CLADE, it will clarify many aspects related to CL discomfort. Also, the authors mentioned the TFOS CLD report in lines 79 – 83. Here some references that could help in elaborate on this: PMID: 29054722, PMID: 29054722, PMID: 29054722.
9. The authors should define the acronym HiSi, which I think that refers to Silicone Hydrogel and should be changed to SiHi.
10. Line 61 – 63: Researchers have recently found that SiHi are related with higher values of lid wiper epitheliopathy in neophytes to CL compared with a control group that have never ever wore CL before. The authors could add some information related to this. PMID: 38712751
11. Line 68: Please add proper reference to statement in this sentence.
12. Line 68 – 70: The authors talked about symptoms in the first paragraph, they should not refer to them in this new paragraph as “this symptomatology” or “these symptoms”. Please reword to “CLADE symptoms” or “CL discomfort”, something similar.
13. Why did the authors select the OSDI questionnaire when exists specific CL discomfort questionnaires such as CLDEQ-8? The authors should define this questionnaire in the introduction section and add any other specific questionnaire if they consider.
14. Line 97: Change “autonomy” to “region” .
15. Line 101: Change “CL’s” to “CLs”.
16. Move line 122 – 123 (informed consent) to the previous paragraph (Line 115 – 119).
17. As I mentioned earlier, the authors should be consistent to how they refer to concepts, i.e. “initial tests”, “baseline”, “first visit”.

Experimental design

1. Inclusion and exclusion should be more precise. Where pregnant or breast-feeding patients included in the study? Did any of the participants have allergies?
2. Line 98 – 102: How was CLADE diagnosed?
3. When the authors state that “All patients kept the same CL” did they refer to the same type/brand or exactly the same CL (1 month perhaps). Line 144 – 145: “regardless of which day of the LC use cycle they were on” So, did some participants start using peroxide with CL in their last days of use and others started with new lenses? Could this affect the current results? Should this variable be controllable by the authors?
4. Which ocular parameters were analysed/collected during the slit-lamp examination? The authors should explain studied ocular parameter, please elaborate on this.

Validity of the findings

1. Categorical variables such as “Bulbar Hyperemia EFRON scale” should be analyzed with cross-tabs and Fisher’s Exact Test.
2. Why did the authors selected periods of 16-years of CL use to create groups?
3. If the authors take some variable to group (i.e. hours per day > 8h and < 8), this variable may not be studied. It is obvious that “daily use” is going to be statistically different between the groups because the variable was used for grouping.
4. Results section is quite chaotic. Maybe the authors should add some subheadings for the different analyses: 1) Intersession and 2) Comparison between each created group.
5. The authors should provide data analyzing differences between baseline and follow-up visits of the entire sample. Did this show any statistical difference between sessions?

Reviewer 3 ·

Basic reporting

1. Indicate in the sample section the distribution of the analysis groups to be used, in order to improve the understanding of the manuscript. Following the sample structure in material and methods, it does not suggest that any differentiation between patients will be made.

2. In the notes to the tables add the corresponding statistical test with the p-value used.

3. In the introduction itself it is mentioned that the use of hydrogen peroxide is already known as a maintenance solution that improves CLADE, it needs to be included more extensively because this study is new.

Experimental design

Lines 98 to 105. To improve the understanding of the manuscript, it would be appreciated if it could be indicated whether all these patients used the same type of LC between them in the sample section. As well as the technical parameters. In the manuscript this information only appears in the results section, but it is of some importance to state it earlier.

Line 99. It is mentioned that the participants had a diagnosis of CLADE, who provided this diagnosis and what classification guide was used for this.

Lines 12-126. The use of the Efron scale after LH scanning is mentioned, but they could indicate which assessments were made or it is assumed that all assessments were made and only bulbar hyperaemia was used for the analysis.

Lines 146 -147. Also in this section it is stated that the eye examination was carried out by different optometrists in the multicentre study. Here the question arises as to whether these are the same optometrists who classify the different parameters after assessment or is this done a posteriori by a masked researcher? If not, this may lead to some bias, so could you explain this section a little more?

Validity of the findings

No comments

---

## Round 0.2 · Minor Revisions

The manuscript has been re-reviewed by the three experts in the field. Minor revisions are necessary before the manuscript is suitable for publication

Reviewer 1 ·

Basic reporting

No comment

Experimental design

Thank you very much for considering my questions and comments and for modifying the manuscript.

Validity of the findings

No comment

Reviewer 2 ·

Basic reporting

I am highly satisfied with the revisions made by the authors to enhance the quality of the manuscript. It is evident that these changes will significantly increase its impact. However, I would like to provide a clarification regarding one of my previous comments.

Experimental design

Everything clear

Validity of the findings

PREVIOUS COMMENT: 1. If the authors take some variable to group (i.e. hours per day > 8h and < 8), this variable may not be studied. It is obvious that “daily use” is going to be statistically different between the groups because the variable was used for grouping.

We are not sure if we understood this comment correctly, but we believe it could be interesting to analyze this variable. Following your example, we group together wearers who wear them correctly for less than 8 hours a day and wearers who do some overuse of contact lenses in excess of 8 hours.
If most wearers above 8 hours stay between 9 and 10 hours and most wearers 8 hours or less are wearing between 7 and 8 hours with this sample size the difference may not be statistically significant if one group averages 8.75±1.2 and the other averages 7.25±1.2 for example. Although it would be unlikely that this would happen, we thought it was important to note it, although, as you say, it may be obvious.

NEW REVIEWER'S REPLY: Another example: If I am studying the ocular parameters of participants with dry eye, and the main parameter used for grouping is BUT, with two groups: 1) BUT < 10 and 2) BUT > 10, I can show the mean and SD of BUT for both groups. However, I can't analyze differences between the groups because there is an inherent bias in the grouping. Therefore, any differences I find between the groups are likely due to this bias.

Reviewer 3 ·

Basic reporting

no comment

Experimental design

Lines 193 to 194, The information about the initial number of participants included in the study. should not go in the results section but in the materials and methods section, as indicated in previous comments.

Line 195-198. In the same way, with the eneral information about the contact lenses they used.

Validity of the findings

no comment

---

## Round 0.3 · accepted · Accept

I have now had the opportunity to read your revised manuscript, and your responses to the reviewers' comments. I believe that you have addressed the concerns raised, and I am happy to accept your manuscript